# Is Bang-Bang Control All You Need?
# Solving Continuous Control with Bernoulli Policies

**Tim Seyde**[1]
MIT CSAIL

**Igor Gilitschenski**
University of Toronto

**Wilko Schwarting**
MIT CSAIL

**Bartolomeo Stellato**
Princeton University

**Martin Riedmiller**
DeepMind

**Markus Wulfmeier**[2]
DeepMind

**Daniela Rus**[2]
MIT CSAIL

## Abstract

Reinforcement learning (RL) for continuous control typically employs distributions whose support covers the entire action space. In this work, we investigate the colloquially known phenomenon that trained agents often prefer actions at the boundaries of that space. We draw theoretical connections to the emergence of bang-bang behavior in optimal control, and provide extensive empirical evaluation across a variety of recent RL algorithms. We replace the normal Gaussian by a Bernoulli distribution that solely considers the extremes along each action dimension - a bang-bang controller. Surprisingly, this achieves state-of-the-art performance on several continuous control benchmarks - in contrast to robotic hardware, where energy and maintenance cost affect controller choices. Since exploration, learning, and the final solution are entangled in RL, we provide additional imitation learning experiments to reduce the impact of exploration on our analysis. Finally, we show that our observations generalize to environments that aim to model real-world challenges and evaluate factors to mitigate the emergence of bang-bang solutions. Our findings emphasise challenges for benchmarking continuous control algorithms, particularly in light of potential real-world applications.[3]

## 1 Introduction

Real-world robotics tasks commonly manifest as control problems over continuous action spaces. When learning to act in such settings, control policies are typically represented as continuous probability distributions that cover all feasible control inputs - often Gaussians. The underlying assumption is that this enables more refined decisions compared to crude policy choices such as discretized controllers, which limit the search space but induce abrupt changes. While switching controls can be undesirable in practice as they may challenge stability and accelerate system wear-down, they are theoretically feasible and even arise as optimal strategies in some settings. It is therefore important to investigate our underlying assumption in designing policies for learning agents, and analyze how deviations from expected behavior can be explained.

Practitioners have empirically observed that even under Gaussian policies extremal switching, or bang-bang control [31], may naturally emerge (e.g. in [21, 41, 51]). Thus, recent work focused on developing methods for preventing this behavior [7] or improving training when bang-bang control is the optimal policy structure [28]. However, understanding the performance, extent, and reasons for emergence of Bang-Bang policies in RL is largely an open research question. Its answer is important for designing future benchmarks and informing empirical and theoretical RL research directions.

---

[1]Correspondence to `tseyde@mit.edu`. [2]Equal advising.

[3]Please find videos and additional details at https://sites.google.com/view/bang-bang-rl

35th Conference on Neural Information Processing Systems (NeurIPS 2021)

In this paper we address these questions in two ways. First, we provide a theoretical intuition for bang-bang behavior in reinforcement learning. This is based on drawing connections to minimum-time problems where bang-bang control is often provably optimal. Second, we perform a set of experiments which optimize controllers via on-policy and off-policy learning as well as model-free and model-based state-of-the-art RL methods. Therein we compare the original algorithms with a slight modification where the Gaussian policy head is replaced with the Bernoulli distribution resulting in a bang-bang controller.

In addition to theoretical justifications, our empirical results confirm emergence of bang-bang policies in standard continuous control benchmarks. Across the board, our experiments also demonstrate high performance of explicitly enforced bang-bang policies: the modified policy head can even outperform the original method. In connection to optimal control theory, we demonstrate how action costs empirically lead to sub-optimal performance with bang-bang controllers. However, we also show the negative impact of action penalties on exploration with Gaussian policies. Due to the necessity of exploration in RL to find the optimal solution, this can result in a complex trade-off.

We also provide a discrete action space version of Maxmimum A Posteriori Policy Optimization (MPO) [1] that avoids relaxations or high variance gradient estimators, which many algorithms rely on when replacing Gaussians with discrete distributions. This is particularly useful as exploration, learning process and final performance are highly entangled in RL, and inaccurate or biased gradients can strongly affect learning. We furthermore aim to mitigate this entanglement by including results for distilling behaviour of a trained Gaussian agent into both a continuous and a discrete agent to compare their performance and provide further evidence for emergent bang-bang behavior.

In summary, our work contains the following key contributions:

1. We show competitive performance of bang-bang control on standard continuous control benchmarks by adapting several recent RL algorithms to leverage Bernoulli policies.

2. We draw theoretical connections to optimal control, motivating the emergence of bang-bang behavior in certain problem formulations even under continuous policy parameterizations.

3. We discuss the introduction of action penalties as a common method to reduce the emergence of bang-bang behaviour, and highlight resulting trade-offs with respect to exploration.

## 2   Related Work

**Policy Representation**   Policies for continuous control are typically chosen to have continuous support and are commonly represented as Gaussians with diagonal covariance matrices [43, 16, 1, 18], Gaussian mixtures [56], Beta distributions [11], or latent variable models [15]. Exploration is then achieved by sampling from the continuous distributions with recent work relating maximum entropy objectives to the need for maintaining sufficient exploration in bounded action spaces [54]. Some approaches consider discretizing the action space to facilitate exploration [14], but need to address the underlying curse of dimensionality [3, 12, 34, 50, 58]. The work by Tang and Agrawal [48] is the most closely related to our work showing that action space discretization can yield state-of-the-art performance for on-policy learning based on a sufficiently large number of discrete components. Here, we provide insight into how even the most extreme discretization of only bang-bang actions can yield state-of-the-art performance for a variety of on-policy and off-policy algorithms.

**Bang-Bang Policies**   Bang-bang control problems have been considered very early on in reinforcement learning [53, 30, 2]. These early works mainly consider problems known to require such a policy type. While it was often observed that bang-bang policies naturally emerge [21, 41, 51] most literature focused on avoiding this behavior [7, 19, 25], while some leveraged it for specific applications [23, 26]. In contrast, our work focuses on understanding its nature, particularly in scenarios that were assumed to require continuous actions for obtaining an optimal controller.

**Relation to Control**   Bang-bang controllers have been extensively investigated in optimal control research. Early works date to the 50s and 60s [45, 4] when it was shown that bang-bang policies arise as optimal controllers in minimum time problems [32, 31]. More recent work in bang-bang control focused on, e.g., further analysis [8, 55], numerical computation [9, 38, 39], and consideration of more specialized systems [10, 29, 35, 57]. Another related line of work are switched dynamical

systems where each mode corresponds to a bang-bang configuration. In these settings, one typically optimizes for switching times given a mode sequence [47] or penalizes the switching frequency as the control effort [46]. Unlike our work, research in this space typically does not consider the dynamics of the learning process in a deep RL setting. However, we can still leverage this existing body of work to inform our explanation for the seemingly surprising emergence of bang-bang policies in RL.

# 3 Optimal Control with Continuous-Time Deterministic Dynamics

We start by describing connections between the emergence of bang-bang behavior in optimal control and common continuous control reinforcement learning problems. First, we characterize three common reward structures together with the optimal controllers they induce. Then, we frame the solutions and their underlying assumptions in the context of our reinforcement learning setting.

We consider a continuous-time deterministic dynamical system with state $s \in \mathbb{R}^n$ and action $a \in \mathbb{R}$. The continuous time-setting simplifies analysis and provides a good approximation under the high sampling rates that are common on continuous control benchmarks and real-world robotic system. We define the non-linear control-affine system dynamics together with the objective function as

$$\dot{s}(t) = f(s(t)) + g(s(t))a(t), \quad 0 \le t \le T, \qquad \text{maximize} \int_0^T r(s(t)) - c(a(t))\mathrm{d}t,$$

where $f : \mathcal{S} \to \mathcal{S}$, $g : \mathcal{S} \to \mathcal{S}$, $t$ is the time, $T$ is the final time, $r : \mathcal{S} \to \mathbb{R}$ is the state reward and $c : \mathcal{A} \to \mathbb{R}$ the action cost. We assume action $a(t)$ to be a scalar in the set $\mathcal{A} = \{a(t) \in \mathbb{R} \mid |a(t)| \le 1\}$. The following derivations can be directly extended to multidimensional inputs [27, Sec. 5.2].

In optimal control settings, this is commonly solved by formulating the corresponding Hamiltonian

$$H(s(t), a(t), p(t)) = r(s(t)) - c(a(t)) + p(t)^T(f(s(t)) + g(s(t))a(t)),$$

where $p(t)$ is the costate variable, and leveraging Pontryagin's maximum principle [6, Prop. 3.3.1][27, Sec. 5.4] to derive the necessary optimality condition for action $a^\star(t) \in \mathcal{A}$ and $0 \le t \le T$ as

$$H(s^\star(t), a^\star(t), p^\star(t)) \ge H(s^\star(t), a(t), p^\star(t)), \quad \forall a(t) \in \mathcal{A}.$$

**Maximum state reward (MS)**  Consider the case when $c(a(t)) = 0$. Based on Pontryagin's maximum principle, the optimal control is determined by solving the following simple linear program

$$\begin{aligned} \text{maximize} \quad & p^\star(t)^T g(s^\star(t))a(t) \\ \text{subject to} \quad & -1 \le a(t) \le 1. \end{aligned}$$

Based on the optimization surface in Figure 1 (left) we see that the optimal action is given by

$$a^\star(t) = \begin{cases} -1 & p^\star(t)^T g(s^\star(t)) < 0 \\ \text{undetermined} & p^\star(t)^T g(s^\star(t)) = 0 \\ 1 & p^\star(t)^T g(s^\star(t)) > 0. \end{cases}$$

If the undetermined condition does not occur, this is referred to as *bang-bang* control.

**Minimum fuel-type cost (MF)**  Consider the case when $c(a(t)) = |a(t)|$. Through a slight reformulation under introduction of optimization variable $z$ we arrive at another simple linear program

$$\begin{aligned} \text{maximize} \quad & z + p^\star(t)^T g(s^\star(t))a(t) \\ \text{subject to} \quad & z \le a(t) \le -z \\ & -1 \le a(t) \le 1. \end{aligned}$$

Based on the optimization surface in Figure 1 (right) we see that the optimal action is given by

$$a^\star(t) = \begin{cases} -1 & p^\star(t)^T g(s^\star(t)) < -1 \\ 0 & |p^\star(t)^T g(s^\star(t))| < 1 \\ 1 & p^\star(t)^T g(s^\star(t)) > 1 \\ \text{undetermined} & |p^\star(t)g(s^\star(t))| = 1. \end{cases}$$

If the undetermined condition does not occur, this yields *bang-off-bang* control (see Appendix A).

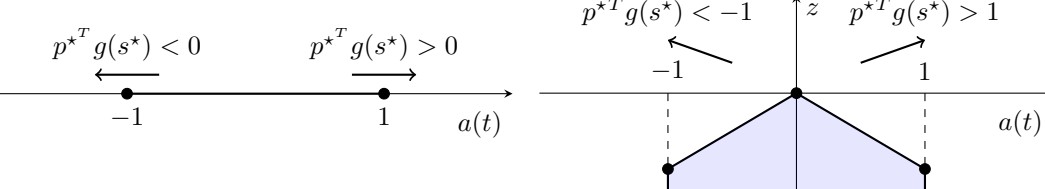

Figure 1: Left: bang-bang solutions with arrows representing the cost function coefficient $p^\star(t)^T g(s^\star(t))$. The feasible region is the line segment $[-1, 1]$. Right: Bang-off-bang solutions with arrows representing the cost function gradients $(1, p^\star(t)^T g(s^\star(t)))$ and feasible region in light blue.

**Minimum energy-type cost (ME)** Consider the case when $c(a(t)) = a(t)^2$. In general, this formulation leads to non bang-bang optimal control [6, Example 3.3.2].

**Singular arcs and chattering behavior** In the undetermined conditions the Hamiltonian is no longer dependent on $a(t)$ and the necessary condition does not allow for determining $a^\star(t)$. These cases are commonly referred to as *singular arcs*. A sufficient condition to avoid singular arcs is having a linear time-invariant dynamical system that is controllable [27, Sec. 5.6]. If an optimal control problem has singular arcs, a bang-bang solution may have to switch infinitely often in a finite time interval to achieve the desired state $s^\star(t)$. This behavior is referred to as *chattering* [37] or *Zeno's phenomenon* [59]. It does not appear in our formulation as we are considering discrete-time dynamical systems with a finite number of discretization points.

**Discretization effect** The Pontryagin maximum principle cannot be directly extended to discrete-time dynamical systems without introducing additional assumptions. For example, Bertsekas [6, Prop. 3.3.2] derives a discrete time version which, in order to provide necessary optimality conditions, requires convexity of the set $\mathcal{A}$. If we consider only bang-bang actions, $\mathcal{A}$ is not convex and the discrete-time derivations no longer hold [6, Example 3.10]. Nevertheless, we find that our discretization intervals are sufficiently small to approximate the continuous case (see also Appendix C).

**Stochastic dynamics** This simplified setup can be seen as a special case of reinforcement learning problems because of its deterministic dynamics. However, similar derivations can be done in case of stochastic action policies and dynamics. For example, this is the case when $\dot{s}(t) = f(s(t)) + g(s(t))a(t) + w(t)$, where $w(t) \in \mathbb{R}^n$ is a random disturbance. Stochastic dynamics, together with the time discretization, are the key components to bridge this setup and the standard RL frameworks, which are discussed in the next section.

**Optimal controller** Solving optimal control problems using the Pontryagin maximum principle is, in general, very challenging since it requires the numerical solution of a set of partial differential equations [6, 27]. Instead, we directly learn switching controllers as stochastic Bernoulli policies.

## 4 Reinforcement Learning Preliminaries

We formulate the control and reinforcement learning problems in the context of a Markov Decision Process (MDP) defined by the tuple $\{\mathcal{S}, \mathcal{A}, \mathcal{T}, \mathcal{R}, \gamma\}$, where $\mathcal{S}$ and $\mathcal{A}$ denote the state and action space, respectively, $\mathcal{T} : \mathcal{S} \times \mathcal{A} \to \mathcal{S}$ represents the density of the transition distribution, $\mathcal{R} : \mathcal{S} \times \mathcal{A} \to \mathbb{R}$ the reward mapping, and $\gamma \in [0, 1)$ is the discount factor. Note that this reward can be obtained from the previous section on continuous time dynamics by defining $R(s_t, a_t) = \int_{\tau=t}^{t+1} r(s(\tau)) - c(a(\tau))\mathrm{d}\tau$. We define $s_t$ and $a_t$ to be the state and action at time $t$, with input constraints of the form $|a_t| \le a_{max}$. Let $\pi_\theta(a|s)$ denote an action distribution parameterized by $\theta$ representing the policy and define the discounted infinite horizon return $G_t = \sum_{t'=t}^{\infty} \gamma^{t'-t} R(s_{t'}, a_{t'})$, where $s_{t+1} \sim \mathcal{T}(s_{t+1}|s_t, a_t)$ and $a_t \sim \pi_\theta(a_t|s_t)$. The objective is to learn the optimal policy maximizing the expected infinite horizon return $\mathbb{E}[G_t]$ under unknown dynamics and reward mappings. This can be achieved by modeling $\pi_\theta(a_t|s_t)$ as a distribution with a neural network predicting the corresponding parameters $\theta$ from $s_t$.

| Domain | Cartpole | Cartpole | Cheetah | Dog | Finger | Humanoid | Quadruped | Walker |
|---|---|---|---|---|---|---|---|---|
| **Task** | Swingup | Sparse | Run | Walk | Spin | Walk | Run | Walk |
| **Type** | MS&ME | MS | MS | MS | MS | MS&ME | MS | MS |

Table 1: DeepMind Control Suite tasks with their objective types based on Section 3.

| Algorithm | Model | Learning | Inputs | Gradient Estimation |
|---|---|---|---|---|
| PPO [43] | model-free | on-policy | states | REINFORCE |
| SAC [16] | model-free | off-policy | states | Reparametrization |
| MPO [1] | model-free | off-policy | states | Expectation Maximisation |
| DreamerV2 [18] | model-based | off-policy | images | REINFORCE & Reparametrization |

Table 2: Algorithms analyzed in conjunction with Bernoulli policies.

## 5 Experiments

In this section, we work to improve our understanding of empirical performance and characteristics of controllers learned via RL in continuous control domains. We compare performance and learnt distributions for Bernoulli and Gaussian policies on several tasks from the DeepMind Control Suite [49] based on a variety of popular RL algorithms. We then analyse the entanglement of exploration and converged solution in RL settings and evaluate robustness of the learned policies. We select MPO for this analysis as it does not require modifications to handle discrete distributions. First, we focus on the final solution by distilling Bernoulli policies from a Gaussian teacher. Next, we discuss advantages and disadvantages of bang-bang behavior for exploration. Finally, we evaluate the effects of augmenting the objective by action penalties on exploration and final solutions of Gaussian and Bernoulli policies to demonstrate the trade-offs when mitigating bang-bang behavior.

### 5.1 Algorithms

We consider Proximal Policy Optimization (PPO) [43], Soft Actor Critic (SAC) [16], Maximum A Posteriori Policy Optimization (MPO) [1] and DreamerV2 [18] as our baseline algorithms. These approaches differ widely in their learning method, optimization strategy, and input signals. A brief comparison of their characteristics is provided in Table 2 with additional description in Appendix B. We investigate the effects of replacing the commonly used Gaussian with a Bernoulli distribution and provide both qualitative and quantitative comparisons of the learned controllers. For approaches requiring reparameterization of Bernoulli policies we leverage the biased straight-through gradient estimator [5] (Appendix B). We place particular focus on MPO as a robust off-policy method which does not require gradient reparameterization and avoids introducing additional estimation errors.

### 5.2 Solving Continuous Control Problems with Bang-Bang Policies

We investigate performance of Bang-Bang policies on several continuous control problems from the DeepMind Control Suite. Figure 2 provides learning curves for PPO, SAC, MPO, and DreamerV2.We note that restricting agents to only select minimum or maximum actions generally does not prevent learning on the tasks and even yields competitive performance. This applies across several domains and distinct algorithm designs (see Table 2).

The presence of action penalties reduces performance of the Bernoulli policies even when the resulting state-space trajectories appear optimal. For instance, bang-bang control is able to swing-up and stabilize on Cartpole Swingup but incurs steady-state cost during stabilization as it is unable to select low-magnitude actions. We also observe lower performance of Bernoulli policies for SAC on Cartpole tasks. SAC presents some unique challenges, as the target entropy parameter operates on different scales in the continuous and discrete distribution cases. We found that particularly domains with low-dimensional action spaces, such as Cartpole, can suffer from premature policy convergence. Finally, bang-bang control with DreamerV2 does not perform well on Finger Spin. The continuous policy already displays some difficulties in learning the task in comparison to PPO, SAC, and MPO. This could indicate that learning a latent model from images can be challenging on this task. Building the model under only extreme input signals potentially exacerbates these effects. Quantitatively, we observe strong performance when restricting the agent to employ Bernoulli policies.

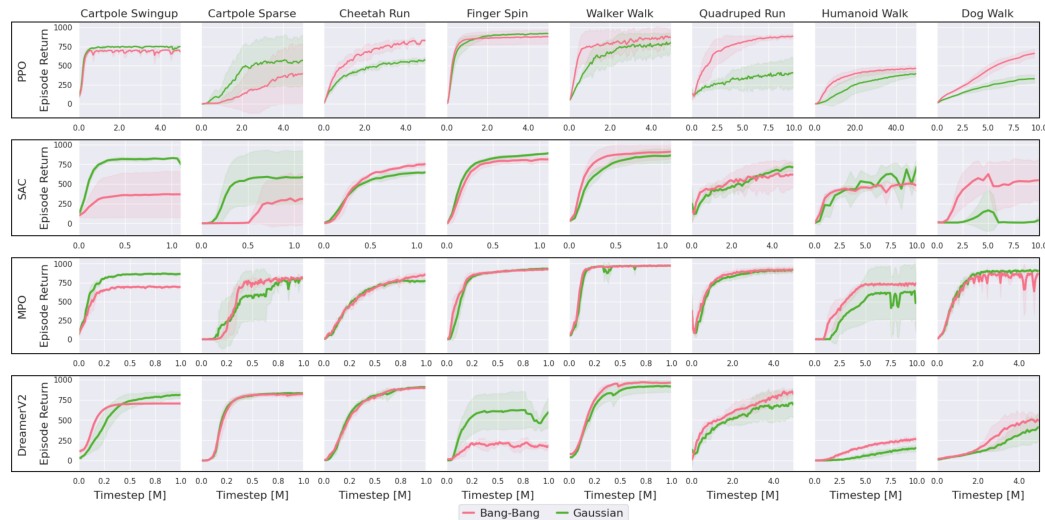

Figure 2: Comparison of Bang-Bang and Gaussian policy heads for solving continuous control tasks. We evaluate PPO, SAC, MPO, and DreamerV2 on several domains from the DeepMind Control suite. Generally, we find that the Bang-Bang policies perform on par with the normal Gaussian policies.

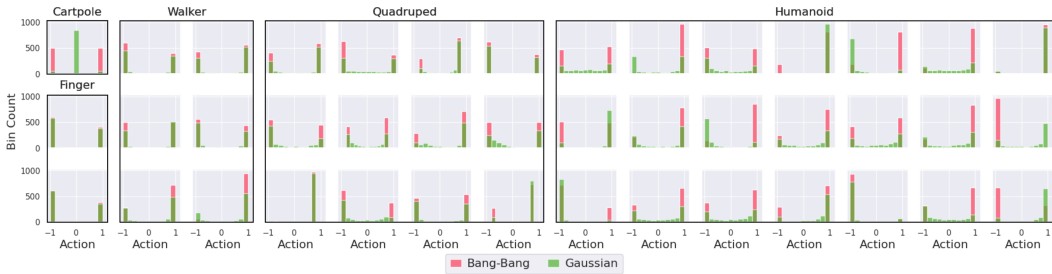

Figure 3: Distribution of actions along a trajectory for MPO. We consider 11 bins per action dimension and aggregate over 1000 steps. The Gaussian policy exhibits bang-bang behavior in several domains. Action penalties, particularly on Cartpole Swingup (top left), can reduce bang-bang action selection.

The results in Figure 2 suggest that across most domains, we do not require a continuous action space. To investigate this phenomenon, we analyze the action distributions of converged policies. Figure 3 provides binned actions aggregated over an evaluation episode for converged Bernoulli and Gaussian policies in several domains. We note that the Gaussian policies consistently resort to extremal actions, particularly on the Finger, Walker and Quadruped tasks. The Cartpole behavior splits into two phases, leveraging bang-bang control for fast swing-up and near-zero actions during stabilization. Qualitatively, we find that the Gaussian policies tend to converge to bang-bang behavior.

### 5.2.1 Disentangling Exploration and Final Solution

The trained Bernoulli and Gaussian policies display strong similarities in their quantitative and qualitative performance. Since the final performance is affected by both exploration and the ability to represent the optimal controller, the Bernoulli policies could compensate for lack of expressivity by enabling better exploration. To focus on the final solution and further investigate similarities between converged policies, we leverage behavioral cloning to investigate the extent to which a Bernoulli policy is able to learn from a Gaussian teacher. The student acts in the environment while being supervised by a converged Gaussian. We maximize log-probability over actions and additionally discretize Gaussian targets for the Bernoulli student. Figure 4 provides learning curves for both Bernoulli and Gaussian students. Strong similarity between curves indicates strong bang-bang characteristics of the Gaussian teacher. The Bernoulli policy performs surprisingly well on all but the

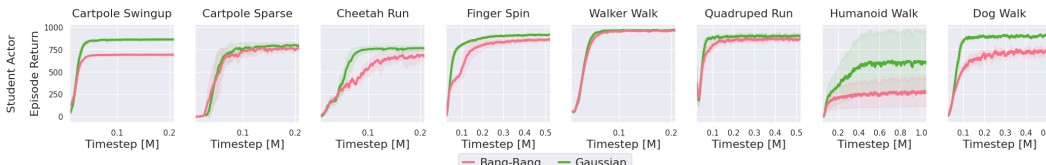

Figure 4: Behavioral cloning with a Gaussian teacher. The Bang-Bang and Gaussian students perform similarly well across several domains, further indicating that the teacher leverages bang-bang control.

Humanoid task. This indicates that the Gaussian teachers consistently leverages bang-bang action selection and its differences from bang-bang behaviour have only limited impact on performance.

## 5.3 Robustness Under Perturbations

We observe similar converged performance of the Bernoulli and Gaussian policies. An argument against bang-bang control for articulated robotic systems from a controls perspective can be stability, particularly in the absence of ideal sensing or actuation delays that may trigger overshooting. We therefore investigate the impact of sensor degradation and modification of the environment parameters based on the Real-World RL Challenge Framework [13].

**Environment Modifications**    To assess whether our observations generalize to more challenging task variations we consider modifications of the environment parameters. Specifically, we investigate performance under changes to the system mass, agent morphology, and friction or damping characteristics across the Cartpole, Walker, and Quadruped domains. Figure 5 (left) provides learning curves for MPO with Bernoulli and Gaussian policies, where parameter values are provided at the top of each subplot in order. We note that both policy types perform comparably across all tasks and parameter values, indicating that the Bang-Bang policy is robust to variations of the environments.

**Transfer under Disturbances**    We evaluate policy transfer under real-world inspired sensor degradation. We consider reduction of the control frequency, stuck sensor signals, dropped sensor signals, sensor delay, and sensor noise (details in Appendix E). The Gaussian action space is a superset of the Bernoulli action space, yielding a richer set of interactions that could translate to better generalization around the optimal trajectory. Figure 5 (right) provides normalized scores relative to performance under ideal sensing based on 40 trajectories. We find that the Bernoulli and Gaussian policies display similar performance. This indicates that robustness to disturbances is not negatively affected by limiting controls to bang-bang within the environments considered. A potential reason for this is that the robot dynamics act as a low-pass filter on the bang-bang input to effectively smooth input jumps.

## 5.4 Characteristics of Bang-Bang Control in RL settings

Learning bang-bang constrained policies can yield competitive performance that generalizes across several environment formulations and algorithm designs. Next, we discuss some characteristics of bang-bang control in RL settings and evaluate methods to avoid such behavior in Gaussian policies.

### 5.4.1 Intrinsic Exploration of Sparse Rewards

**Pointmass**    Bang-bang policies only apply extremal actions. By forcing highest magnitude actions, the intrinsic exploration characteristics of an agent are affected. Figure 6 provides two tasks for a point-mass controlled in x-y position. Each task is evaluated with no or quadratic action penalties (left, right). Here, we consider Bang-Bang, Bang-Off-Bang, and Gaussian policies. In the top row, the agent is always initialized at the center and does not receive any rewards. The Bang-Bang policy exhibits the largest area coverage by virtue of only sampling large magnitude actions. This effect is magnified under action cost, as other agents limit exploration to reduce cost (see also Appendix F). In the bottom row, the agent starts at the center without resets and receives sparse rewards at the circles. The Bang-Bang policy again exhibits the largest coverage. While its large magnitude sampling facilitates escapes from local optima, it can also impact stabilization at global optima (Figure 6, right). Overall, Bang-Bang policies offers strong intrinsic exploration that is robust to action penalties.

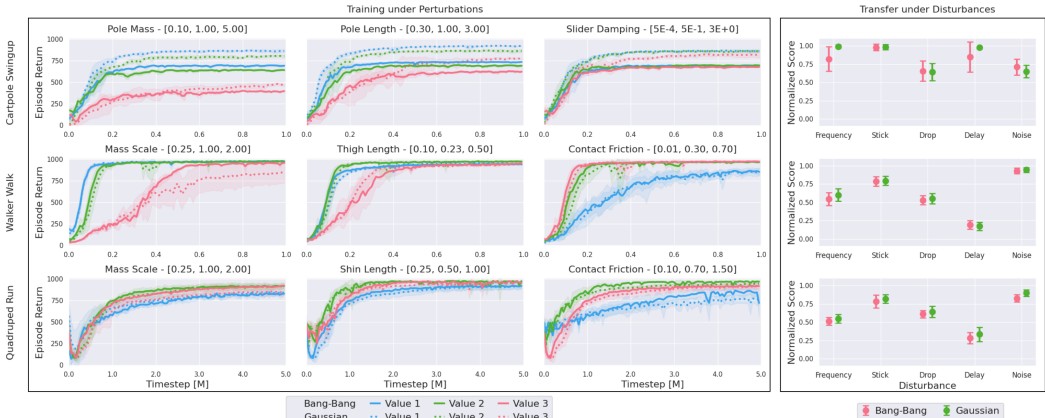

Figure 5: Robustness of the Bang-Bang policies when considering task modifications based on the Real-World RL suite [13]. Left: training with perturbed environment parameters. Each subplot compares the effect of changing one parameter on training performance. Right: transfer under disturbances, performance normalized with respect to undisturbed scenario. Bang-Bang and Gaussian policies perform comparably, indicating similar robustness in simulation.

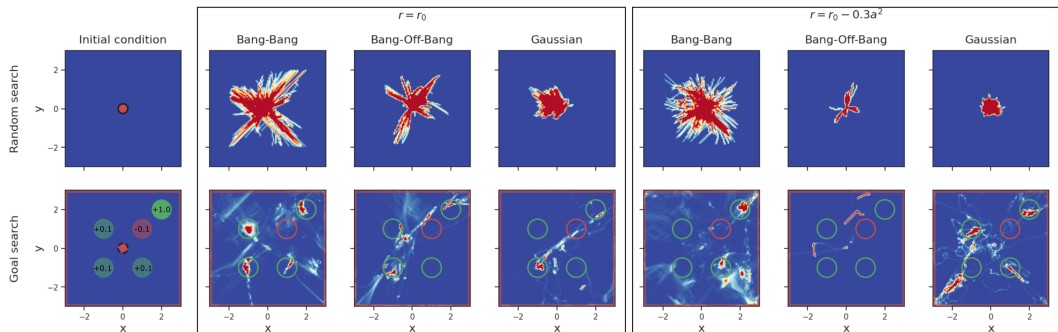

Figure 6: Pointmass exploration tasks. Top: the agent is reset every episode with no reward feedback. Bottom: the agent always continues and receives different sparse rewards as indicated by the values in the circles. Bang-Bang control leverages extreme actions yielding strong passive exploration, unaffected by action penalties (top). This can facilitate escape from local optima and impede stabilization at global optima (bottom). In sparse exploration tasks, action penalties may prevent the agent from sufficient exploration when it is able to focus on minimizing action cost instead.

**Control Suite** While action penalties limit maximum performance of Bang-Bang policies, they can also negatively affect exploration in Gaussians as observed in Figure 6. Exploration is hindered whenever reward maximization is traded-off to minimize action cost. This behavior is favored by sparse feedback, as the agent needs to actively explore to observe rewards. We compare performance of Bang-Bang, Bang-Off-Bang, and Gaussian policies on three versions of the same tasks: dense, sparse, and sparse with action cost (details on reward sparsification are provided in Appendix D). Figure 7 provides learning curves for the Cartpole Swingup, Walker Walk, and Quadruped Run domains. All three policy parameterizations perform similarly on the dense versions of the tasks. The sparse Cartpole and Walker tasks without action cost yield similar converged performance for all policies, while the Bang-Bang policy converges faster due to its strong passive exploration. On the sparse tasks with action penalties, both the Gaussian and Bang-Off-Bang controllers are unable to solve the task and focus on action cost minimization instead of exploring sufficiently in search of reward (convergence to 0 returns). The Bang-Bang controller solves these tasks with steady-state action cost, as it only selects maximum magnitude actions and effectively ignores the detrimental impact of action penalties on exploration. This example further highlights the intricate interplay between finding policies that avoid bang-bang behavior and enabling sufficient exploration.

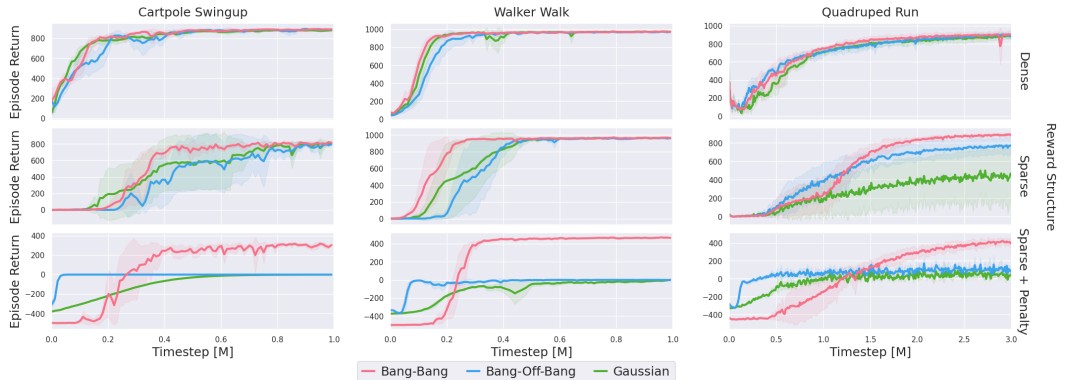

Figure 7: Exploration under sparse reward feedback and action penalties for Bang-Bang, Bang-Off-Bang, and Gaussian policies. While all three policy types perform comparably on the dense task versions (column 1), Bang-Bang policies can yield faster convergence on sparse tasks due to passive exploration arsing from extremal action selection (column 2). Furthermore, large action penalties can limit exploration in favor of action cost reduction for the non-Bang-Bang policies (column 3).

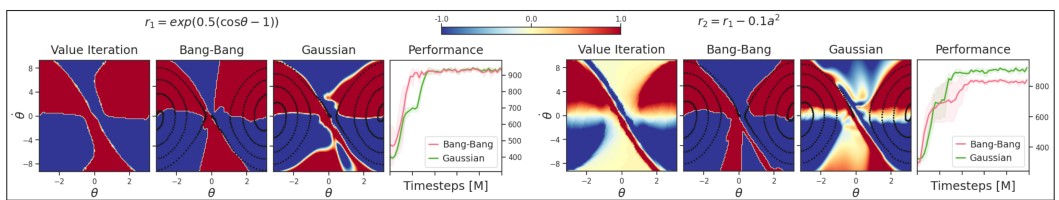

Figure 8: Learning to swing-up a pendulum under no action cost (left) and quadratic cost (right). The Bang-Bang policy converges faster without action penalties, the Gaussian yields higher returns with action penalties. Learned policy mappings are overlayed with sample swing-up trajectories (black).

### 5.4.2 Action Penalties

Bang-Bang policies enable fast exploration and can arise as optimal controllers in the absence of action cost. However, robotics applications may suffer from extensive extremal action selection and action penalties can mitigate bang-bang behavior. We highlight these considerations on a pendulum swing-up task with no and quadratic action cost in Figure 8. Without action cost (left), both Bang-Bang and Gaussian policies learn to approximate the optimal action mapping and solve the task with the Bang-Bang policy offering a faster rise time. With action cost (right), representing the minimum energy category in optimal control (Section 3), the Bang-Bang policy is unable to represent the optimal action mapping and continues to incur steady-state cost (see also Appendix F).

We investigate the effect of augmenting the reward function of the more complex Walker and Quadruped tasks with action penalties. Here, we consider both quadratic cost to reduce action magnitude and difference cost to increase smoothness. In the latter case, we augment the system state by the previous action. Figure 9 (top) displays Gaussian means along a Quadruped Run trajectory for the different penalty structures and their combination. We observe the desired effect of lower magnitude and smoother transitions in the learned distributions. Figure 9 (bottom) provides training curves under the different reward structures, performance based on the nominal reward function, and transfer under disturbances (see Section 5.3) for the Walker (left) and Quadruped (right) tasks. We note that while action penalties help to mitigate bang-bang behavior, robustness of the resulting gaits is only slightly increased. Furthermore, the modified reward structure may yield reduced performance of trained agents as measured by the nominal reward function and observed on the Quadruped. These findings suggest that while potentially undesired bang-bang action selection can be reduced by introducing action penalties, naive engineering of the reward structure may force the agent into sub-optimal behaviors. The question then becomes what metrics to consider when evaluating agent performance in light of transfer to real-world systems. Pure return maximization may exploit both simulation and task peculiarities and is not sufficient for comparing the merit of different algorithms.

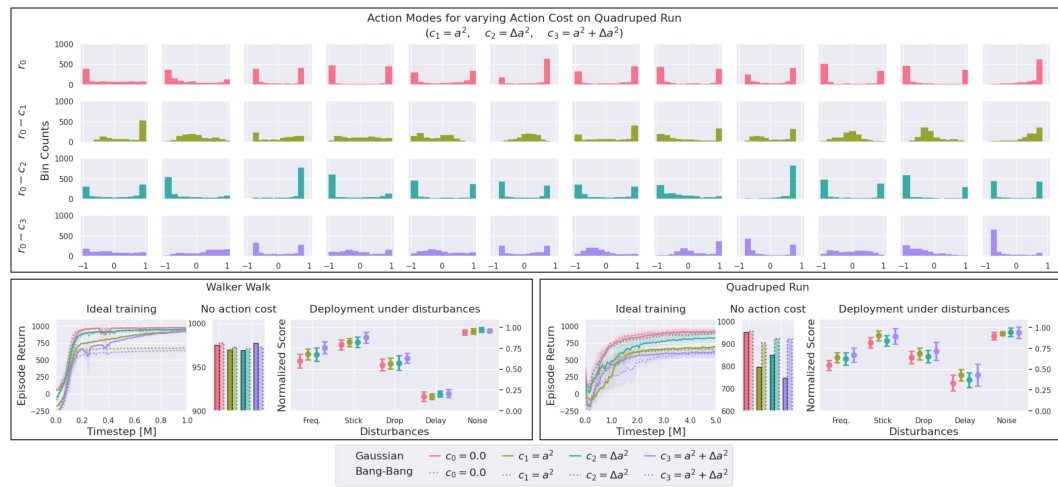

Figure 9: Impact of action penalties. Top: Gaussian means under varying penalties along a Quadruped Run trajectory. Bottom: training and transfer performance on Walker (left) and Quadruped (right). While action penalties help to mitigate bang-bang behavior, the resulting gaits only slightly increase robustness and may reduce performance as measured by the nominal reward function (Quadruped).

## 6 Discussion

In this paper, we investigate the emergence of bang-bang behavior in continuous control reinforcement learning and show that several common benchmarks can in fact be solved by bang-bang constrained Bernoulli policies. We draw theoretical connections to bang-bang solutions from optimal control and perform extensive experimental evaluations across a variety of state-of-the-art RL algorithms. In particular, we theoretically derive when bang-bang control emerges for continuous-time systems with deterministic dynamics and describe connections to continuous control RL problems considered here.

In our experiments, we find that many continuous control benchmarks do not require continuous action spaces and can be solved (close to) optimally with learned bang-bang controllers for several recent model-based and model-free, on-policy and off-policy algorithms. We further demonstrate the efficiency of bang-bang control in more realistic variations of common domains based on the Real-World RL suite [13] and demonstrate how action costs affect the optimal controller. Generally, Bernoulli policies may be applicable when the dynamics act as a low-pass filter to sufficiently smoothen the switching behavior. A key advantage can then be the reduction of the policy search space from $\mathcal{R}^N$ to $2^N$ with $N = |\mathcal{A}|$, while the resulting extreme actions can favor coarse exploration. Since exploration, learning and the final solution are entangled in RL, we perform additional experiments with focus on behavioural cloning of a trained Gaussian controller to compare the performance of bang-bang and continuous policies without the additional challenges arising from exploration. It is important to note that we still require sampling-based optimisation, which has become popular to handle continuous action spaces. However, the main reason is often not the continuous action space but the potential high-dimensionality of the action space which would otherwise become intractable.

Benchmark design is a complex task with the goal of measuring progress that is transferable to impactful real-world applications. We demonstrate that the learnt controllers for continuous control benchmarks without action penalties can exhibit bang-bang properties which can be damaging to real-world systems. At the same time, we also show that including action penalties can significantly impact exploration for Gaussian policies. Integrating the impact of task design on both exploration as well as final solution properties and investigating algorithms which can overcome local optima while still learning smooth controllers is an important direction for future work.

**Societal Implications:** Improved understanding of emergent bang-bang behavior may broaden real-world applicability of RL policies. While resulting approaches could be used in ways not intended by the researchers there are many beneficial applications such as robotics for improved productivity and workplace safety. Even desirable outcomes can come with side-effects such as job loss and societal transformation costs. While the authors ultimately believe that societal benefits of this work outweigh its harms, these considerations need to be re-evaluated on a constant basis as new applications emerge.

## Acknowledgments and Disclosure of Funding

Tim Seyde, Igor Gilitschenski, Wilko Schwarting and Daniela Rus were supported in part by the Office of Naval Research (ONR) Grant N00014-18-1-2830 and Qualcomm. This article solely reflects the opinions and conclusions of its authors and not any other entity. We thank them for their support. The authors further would like to thank Lucas Liebenwein for assistance with cluster deployment, and acknowledge the MIT SuperCloud and Lincoln Laboratory Supercomputing Center for providing HPC resources. We would also like to thank Murad Abu-Khalaf for insightful discussions as well as the NeurIPS reviewers and program chairs for their helpful feedback and suggestions for improvement.

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
