## A  Minimum-fuel cost (MF) derivation

When $c(a(t)) = |a(t)|$, from Pontryagin's maximum principle [27, Sec. 5.5], the necessary optimality condition for action $a^\star(t) \in \mathcal{A}$ and $0 \leq t \leq T$ is

$$H(s^\star(t), a^\star(t), p^\star(t)) \geq H(s^\star(t), a(t), p^\star(t)), \quad \forall a(t) \in \mathcal{A},$$

$$\text{therefore,} \quad -|a^\star(t)| + p^\star(t)^T g(s^\star(t)) a^\star(t) \geq -|a(t)| + p^\star(t)^T g(s^\star(t)) a(t), \quad \forall a(t) \in \mathcal{A}.$$

To maximize the right-hand side, $a(t)$ must solve the following linear optimization problem

$$
\begin{array}{ll}
\text{maximize} & z + p^\star(t)^T g(s^\star(t)) a(t) \\
\text{subject to} & z \leq a(t) \leq -z \\
& -1 \leq a(t) \leq 1,
\end{array}
\tag{1}
$$

where $a(t)$ and $z$ are the optimization variables. Figure 1 on the right shows the feasible region of (1) and the corresponding cost function gradient $(p^\star(t)^T g(s^\star(t)), 1)$. If $p^\star(t)^T g(s^\star(t)) < -1$, the solution corresponds to vertex $(a^\star(t), z^\star) = (-1, -1)$, while, if $p^\star(t)^T g(s^\star(t)) > 1$, the solution ends up at vertex $(a^\star(t), z^\star) = (1, -1)$. In case $-1 < p^\star(t)^T g(s^\star(t)) < 1$, the maximum is attained at $(a^\star(t), z^\star) = (0, 0)$. Finally, if $p^\star(t)^T g(s^\star(t)) = 1$, any $0 \leq a(t) \leq 1$ and $z = -a(t)$ achieve the maximum. Similarly, if $p^\star(t)^T g(s^\star(t)) = -1$, any $-1 \leq a(t) \leq 0$ and $z = a(t)$ achieve the maximum. The last two cases correspond to cost function gradients perpendicular to the faces of the feasible region. In linear optimization, these cases give is an infinite number of optimal solutions between two adjacent vertices.

## B  Baseline Algorithms and Gradient Estimation

### B.1  Baseline Algorithms

We provide a brief discussion of the baseline algorithms below. The libraries our implementations are based off for PPO, SAC, and DreamerV2 are available under the MIT License, and the base MPO implementation under the Apache License 2.0. For MuJoCo [52], we used a Pro Lab license.

**PPO**  Proximal Policy Optimization [43] is a model-free on-policy algorithm. It aims to maximize expected improvement while guarding against policy collapse by limiting the policy update magnitude. Here, we consider PPO with a clipped surrogate objective and early stopping based on the mean KL-divergence from the previous policy. Our implementation is based on the Tonic library [42].

**SAC**  Soft Actor Critic [16] is a model-free off-policy algorithm. It aims to maximize expected improvement on an entropy-regularized objective and implicitly trades off exploration with exploitation. Here, we optimize the entropy coefficient $\alpha$ and leverage the clipped double-Q trick to stabilize learning. We build on the SAC implementation provided by the Softlearning library [17].

**MPO**  Maximum a Posteriori Policy Optimization [1] is a model-free off-policy algorithm. It alternates between a KL-regularized optimization of a non-parametric policy on samples from the state-action value function and fitting a parametric policy to this non-parametric target. We leverage decoupled KL-constraints in optimizing the parametric policy and employ Retrace [40] for learning the state-action value function. We extend the implementation provided by the Acme library [20].

**DreamerV2**  DreamerV2 [18] is a model-based off-policy algorithm. It learns a recurrent latent variable model based on visual inputs and optimizes the policy on entropy-regularized $\lambda$-returns from model rollouts. We follow the original authors in parameterizing the baseline policy as a diagonal truncated Gaussian for continuous control. Our implementation builds on the DreamerV2 codebase.

**Bijectors**  The underlying distributions are mapped to the environments' action spaces via a bijector. This consist of a shift and scale operations for Categorical and Gaussian policies, while the latter may additionally leverage a tanh bijector.

**Algorithm 1** Reparameterization gradient for Gaussian

```
1 mean, scale = network(inputs)              # conditional moments
2 sample = gaussian(0.0, 1.0)                # sample without gradient
3 sample = mean + scale * sample            # sample with moments gradient
4 action = bijector(sample)                  # transform to action space
```

**Algorithm 2** Straight-through gradient for Categorical

```
1 probs = network(inputs)                    # conditional probabilities
2 sample = one_hot(probs)                    # sample without gradient
3 sample = sample + probs - no_gradient(probs)  # sample with probs gradient
4 action = bijector(sample)                  # transform to action space
```

### B.2 Gradient Estimation for Stochastic Computation Graphs

Random sampling operations are generally not differentiable. To enable backpropagation through stochastic policies, we leverage the common reparameteriztion trick for Gaussian policies in Algorithm 1. For Categorical policies, we apply the biased straight-through gradient estimator [5] in Algorithm 2. We also tried the Gumbel-Softmax estimator [22, 36], but did not find this to improve performance. The optimization procedure employed by MPO bypasses the need for gradient estimation via reparameterization, which particularly reduces bias when optimizing Categorical policies.

## C   Training Details

Experiments were each conducted on 4 CPU cores in combination with 1 GPU (NVIDIA V100). All reported means and standard deviations are based on 4 runs differing in their random seeds. To estimate transfer performance, we computed mean and standard deviation across 5 runs per seed for a total of 20 trajectories as disturbances are probabilistic. Throughout, we consider episodes with the standard length of 1000 timesteps. For the experiments that down-sample the control frequency, we kept the episode duration constant and adapted the number of steps per episode accordingly. Throughout, we use the default parameters from each algorithm codebase. For MPO, we bound the non-Gaussian policy parameters in the decoupled KL constraint by $\epsilon = 0.1$. For PPO, we found that learning distinct scale parameters per action dimension improves performance for Gaussian policies. The environments considered here operate at a default control frequency of 40Hz to 100Hz, while real-world systems may even run the control loop at frequencies above 100Hz [24, 33].

## D   Sparse environments

**Cartpole**   The two sparse versions use the original Cartpole Swigup Sparse task. The dense version is constructed by taking the original Cartpole Swingup task and removing both action and velocity penalties from the reward function to create a dense version of the sparse task.

**Walker**   The dense version uses the original Walker Walk task. The sparse versions are created by thresholding step rewards at $r_{th} = 0.5$, setting all rewards below to 0 and re-mapping rewards above to [0, 1], as in [44]. The sparse reward is given by $r_{sparse} = \text{clip}(r_{dense} - r_{th}, 0, 1 - r_{th})/(1 - r_{th})$.

**Quadruped**   The dense and sparse task versions follow the procedure outlined for Walker Walk. For the Gaussian policy on the Sparse + Penalty version we consider only 3 seeds as one run terminated.

**Action Penalty**   We use a quadratic action cost (ME type) to generate the reward $\hat{r} = r - 0.5a^2/|\mathcal{A}|$.

| Disturbance | Control Frequency | Obs. Stuck | Obs. Drop | Obs. Delay | Obs. Noise |
|---|---|---|---|---|---|
| **Parameters** | Cart. ; Walk. ; Quad. | Prob. ; Steps | Prob. ; Steps | Steps | Std. Dev. |
| **Value** | $\times 0.1$ ; $\times 0.2$ ; $\times 0.25$ | 0.05 ; 5 | 0.05 ; 5 | 6 | 0.3 |

Table 3: Disturbances considered to evaluate transfer robustness in Sections 5.3 & 5.4.2.

# E  Disturbance Parameters

The experiments on transfer robustness in Sections 5.3 & 5.4.2 use the disturbance parameters in Table 3. The control frequency disturbance down-samples the control by the value indicated for the Cartpole, Walker, and Quadruped domains. For the observation disturbances, we selected the medium disturbances from the Real-World RL Challenge framework [13]. The Stuck sensor disturbance does not update a sensor reading for several timesteps, while the Dropped sensor disturbance zeros a sensor reading for several timesteps. Both disturbances are probabilistic, taking effect with probability 0.05 and lasting for 5 timesteps. The observation delay shifts all observation by 6 timesteps, while the observation noise applies additive white Gaussian noise with standard deviation 0.3.

# F  Additional Experiments

We provide additional empirical insights into how the reward structure, in conjunction with action costs, affects both optimal policy parameterization and exploration. Additionally, we evaluate benchmark performance of Bang-Off-Bang policies for MPO and show the distribution of actions for converged Gaussian policies along trajectories on additional seeds.

## F.1  Optimal Policy under Varying Action Cost

We expand our study of optimal policy design for the pendulum swing-up task in Figure 8. In addition to the minimum state (MS) and minimum energy (ME) reward structures, we further consider minimum fuel (MF) type rewards through introduction of an absolute value action penalty. Figure 10 provides learned policy mappings and performance curves for Bang-Bang, Bang-Off-Bang, and Gaussian policies. The action cost differs by row, where the corresponding optimal policy parameterization is highlighted in green. While the Gaussian can represent the optimal policy in each scenario, its increased expressiveness can require more samples for convergence. This can be observed for the MS reward structure, where Bang-Bang control is optimal (top row, right). Similarly, Bang-Off-Bang policies can suffer from slower exploration due to the high probability of choosing 0 actions (see also Section 5.4, Figure 6). This can be observed for the MF reward structure, where the Gaussian policy converges much faster than the Bang-Off-Bang policy (middle row, right). Generally, the Bang-Bang policy enables fast convergence due to its passive exploration (Section 5.4, Figure 6), but incurs steady-state cost when the objective includes action penalties.

## F.2  Benchmarking Bang-Off-Bang Control

We provide additional benchmarking results for Bang-Off-Bang policy heads on MPO in Figure 11. Bang-Bang, Bang-Off-Bang, and Gaussian policies perform similarly across most tasks considered. Bang-Bang policies converge faster than Bang-Off-Bang policies due to their exclusion of 0 actions which forces stronger passive exploration (e.g. Cartpole Sparse, Walker Walk). However, Bang-Off-Bang policies can leverage 0 actions to avoid steady-state action cost (e.g. Cartpole Swingup).

## F.3  Additional Action Distribution Data

We provide distributions of aggregated actions along trajectories for additional seeds in Figure 12. On tasks without action penalties, the Gaussian policies tend to exhibit strong bang-bang behavior sampling actions close to the minimum or maximum bounds (Finger, Walker, Quadruped). Inclusion of action penalties can reduce emergence of bang-bang solutions, while introducing additional trade-offs with respect to performance on the original objective and exploration capabilities (Sections 5.4.2 & 5.4.1). On Cartpole Swingup, which includes action cost, the Gaussian agent still uses bang-bang action selection during swing-up and only switches to low-magnitude actions for stabilization at the

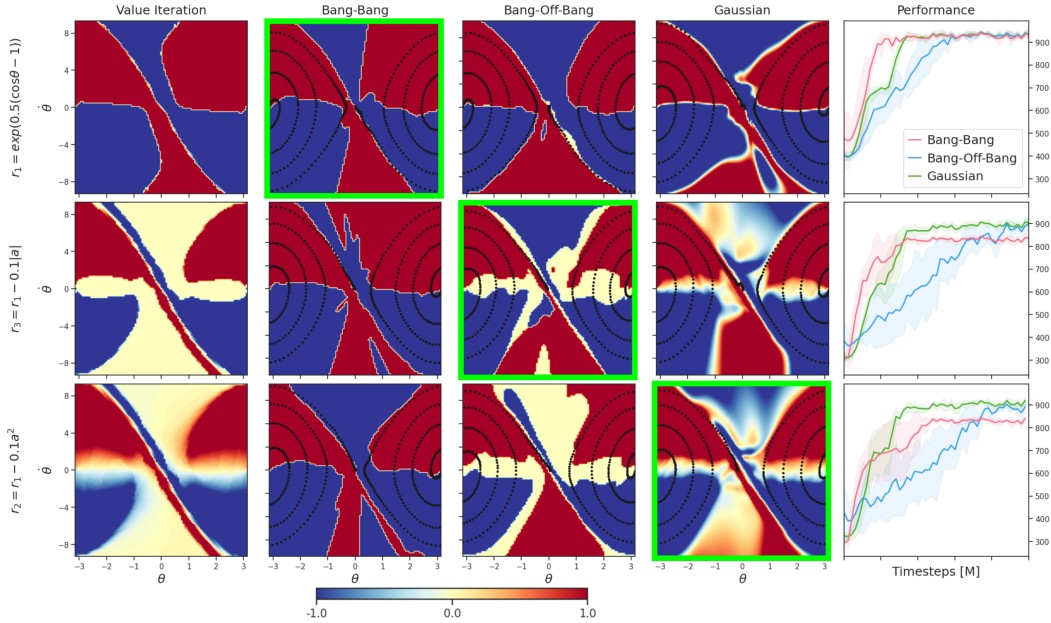

Figure 10: Learning to swing-up a pendulum under varying action cost. Top to bottom: dense reward function without action cost, with absolute value action cost, and with squared action cost. Columns 1-4 provide the policy mapping computed via Value Iteration and learned via Bang-Bang, Bang-Off-Bang and Gaussian policies, respectively. Column 5 compares performance. The optimal policy parameterization for each action cost (row) is highlighted (green). In RL problems, the ideal parameterization might not yield the best learning dynamics (Bang-Off-Bang vs. Gaussian, row 2).

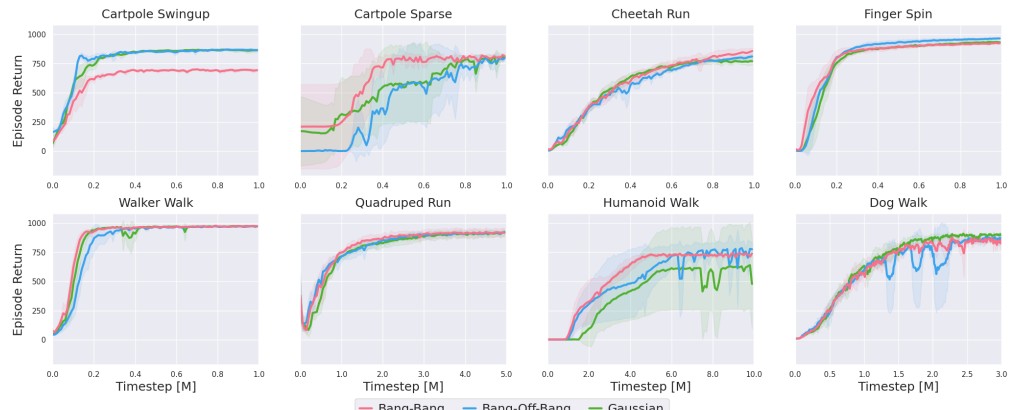

Figure 11: Comparison of Bang-Bang, Bang-Off-Bang, and Gaussian policy heads for MPO on continuous control tasks. The policies perform similarly on most tasks, while the Bang-Bang policy can incur steady-state action cost (e.g. Cartpole Swingup). The Bang-Off-Bang policy leverages the 0 action to avoid this steady-state penalty at the risk of reducing convergence speed (Cartpole tasks).

top. On Humanoid Walk, emergence of bang-bang behavior differs both across action dimensions and seeds. This further underlines that static action penalties do not necessarily guard against local optima that can solve a task sufficiently well by relying, at least in part, on large control switches.

## F.4 Action Penalties

Figure 13 provides an enlarged version of Figure 9 for improved readability.

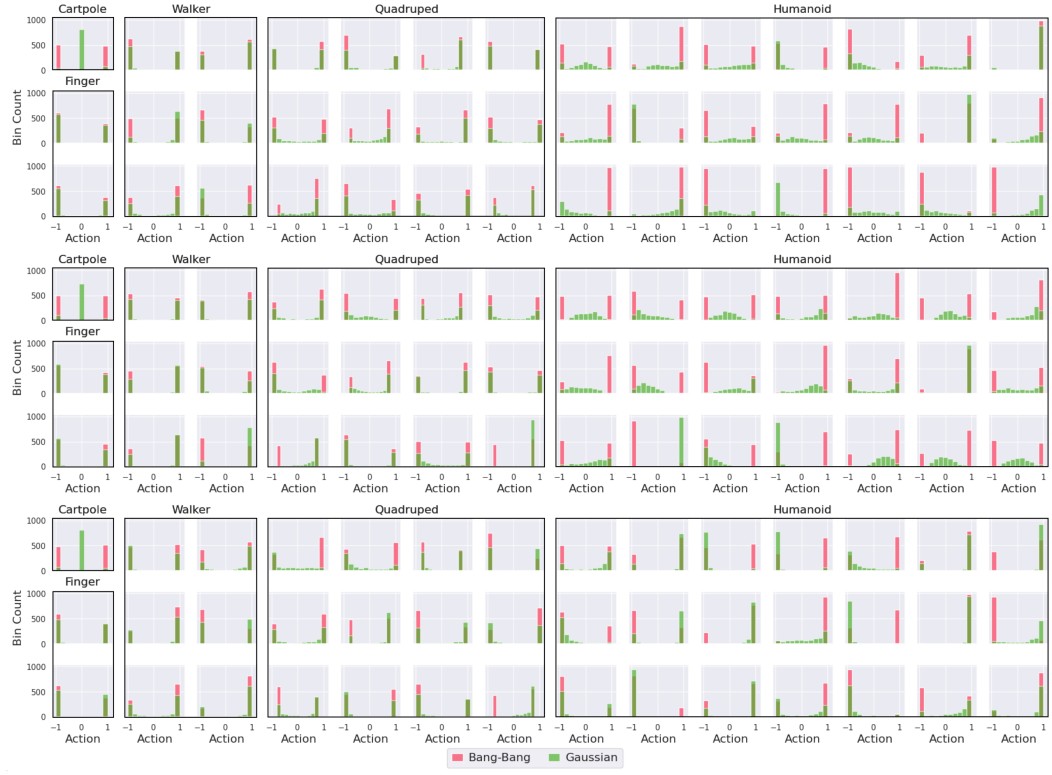

Figure 12: Distribution of actions for MPO on additional seeds. We consider 11 bins per action dimension and aggregate over 1000 steps. The Gaussian policy exhibits bang-bang behavior in several domains, while presence of action penalties on Cartpole and Humanoid reduces this to some extent.

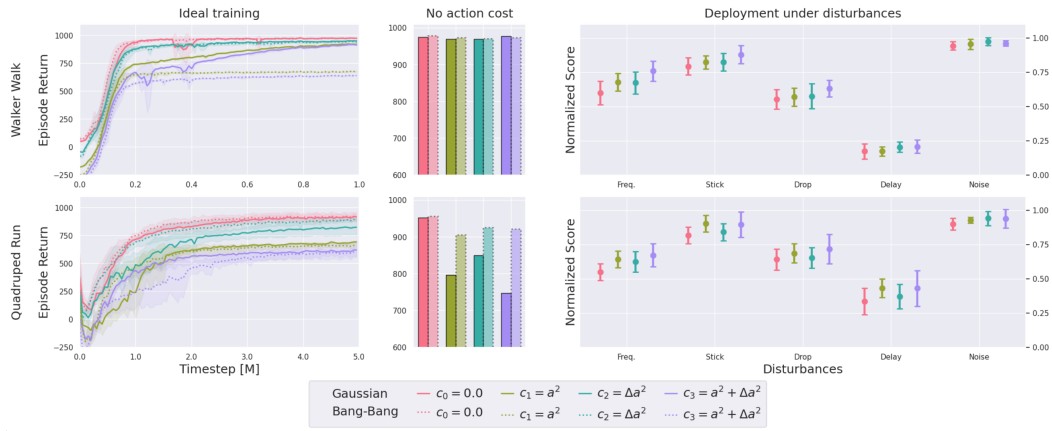

Figure 13: Training and transfer performance on Walker (top) and Quadruped (bottom). While action penalties help to mitigate bang-bang behavior, the resulting gaits only slightly increase robustness and may reduce performance as measured by the nominal reward function (Quadruped).