# OpenReview forum: "Is Bang-Bang Control All You Need? Solving Continuous Control with Bernoulli Policies"
_NeurIPS.cc/2021/Conference — NeurIPS 2021 Poster_

### Official Review · Reviewer_39gS · 2021-07-14

**Rating:** 6
**Confidence:** 3

**Summary:**

The paper focuses on the so-called bang-bang policies. These are ones, which are allowed only for discrete values, typically two extremal ones. Even though they are usually disadvantageous in real-world scenarios, they tend to arise in simulation, and they seem to deliver at least decent performance.

**Limitations And Societal Impact:**

I am not satisfied with the limitation description. Generally, I am fond of merging this with 'the further research discussion' (as the authors did). However, in this case, it is blurred to the point, I do not know what the authors think about the limitations of this paper.

**Main Review:**

The paper focuses on the so-called bang-bang policies. These are ones, which are allowed only for discrete values, typically two extremal ones. Even though they are usually disadvantageous in real-world scenarios, they tend to arise in simulation, and they seem to deliver at least decent performance. I find the paper overall interesting and valuable; it is all generally well written. It presents a theoretical analysis and proper empirical evaluation. Below I list the pros and cons and some (typically minor) recommendations and questions.



Pros:

* Choices of algorithms covers well the spectrum of the current state-of-the-art
* The authors attempt to make a deeper analysis than just observing the basic phenomenon
* Related work seems fairly complete (I am not an expert though, see also the cons section)
* The paper is overall well-structured, well-written and easy to follow (with exceptions mentioned in the cons section)
* In general, I appreciate additional experiments



Cons:

* The papers fall into a benchmark-like category. As such, there are higher expectations as to the empirical section. I realise that it is always a hard choice, and one needs to stop. Still, I'd appreciate having more environments. Moreover, the authors use $4$ seeds, which might be too small for RL settings.
* I find Figure 1 somewhat overwhelming. I appreciate putting all the training data. However, I'd find a useful summary.
* I am not quite convinced about the exploration section. It is almost by design that bang-bang spans a bigger region (which the authors mention). If the conclusion is that this is an issue for exploration, I'd appreciate algorithmic developments towards increasing exploration in the Gaussian setting.
* Similar problems, though from a different perspective, were studied in https://arxiv.org/pdf/1910.02208.pdf. I'd suggest the authors should at least mention this in Related Word or, better, compare SOP with bang-bang SOP.
* I find Action Penalty somewhat incomplete and incomprehensive (again, I am overwhelmed by Fig 7. )
* (Nitpick) The authors do not specify technical details of the environments. It'd be nice to make the paper self contain.
* (Nitpick) I'd appreciate more details about discretised SAC than Alg 2 (say what is bijector)
* (Nitpick) It is perhaps good to mention that discretisation was superior in a popular Learning to run contest: "https://arxiv.org/abs/1804.00361"



Questions and suggestions:

* I acknowledge the theoretical section. I'd appreciate some discussion on how much the studied model is relevant to the environments in the paper and real-world scenarios.
* (Nitpick) I'd appreciate having more hypotheses, why bang-bang work so well.



Originality: The paper is mostly a benchmark paper.

Quality: Yes, this is a fair empirical work with some theoretical background.

Clarity: The paper is clear.

Significance: Comprehensive study of the bang-bang policies presented in the submission can be beneficial for the community.

**Time Spent Reviewing:**

8h

---

> ### Author Response · Authors · 2021-08-11
> **Response to Reviewer 39gS**
>
> We would like to thank the reviewer for their time and constructive feedback. We are very happy that they recognize our study’s value to the RL research community for informing future algorithm designs. In the following, we will try to clarify individual open questions and comments.
>
> >The papers fall into a benchmark-like category. [...]  I'd appreciate having more environments. Moreover, the authors use 4 seeds, which might be too small for RL settings.
>
> We agree that the paper has benchmarking characteristics. Our focus is less on detailed numeric comparisons for every possible domain, and we instead aim to confirm general trends across several recent RL algorithms with very different properties (e.g. on-/off-policy, model-free/-based). Running on 4 seeds is close to the standard 5 seeds (some approaches use 3 seeds), while replication across 4 algorithms yields a total of 16 distinct runs per environment. We evaluate on tasks that are commonly used in the literature to report SOTA performance (DeepMind Control Suite), while extending analysis to benchmarks that aim for more realistic learning (Real-World RL Suite). Throughout, our main goal is to highlight unexpected trends in continuous control RL and point towards potential challenges and shortcomings in evaluating Gaussian agents.
>
> >I'd appreciate some discussion on how much the studied model is relevant to the environments in the paper and real-world scenarios [...] [as well as] more hypotheses, why bang-bang works so well.
>
>
> Our goal is not to advocate for the use of Bernoulli policies. Instead, we show that certain problem formulations encourage bang-bang behavior, and how Bernoulli policies can yield stable learning to reach competitive performance on common continuous control benchmarks. Generally, Bernoulli policies can be applicable to systems where the dynamics act as a low-pass filter to smoothen the switching behavior (see also Pulse-width modulation). However, strong switching inputs can destabilize control systems and are not well suited for e.g. generating position targets for PD tracking control. If the system is compatible with bang-bang inputs, a key advantage can be the reduction of the policy search space from R^N to 2^N (|A| = N). Furthermore, the resulting extreme actions can favor coarse exploration where Gaussians may sample smaller changes in state between timesteps. We will expand our discussion to highlight these aspects.
>
>
> >I am not satisfied with the limitation description. Generally, I am fond of merging this with 'the further research discussion' (as the authors did). However, in this case, it is blurred to the point, I do not know what the authors think about the limitations of this paper.
>
> We show that several recent RL algorithms do not require continuous or even fine-grained discrete policy distributions for reaching (near-) SOTA performance on common continuous control benchmarks. While our observations are limited to the algorithms and benchmarks considered, they cover a significant part of current RL research efforts and evaluation metrics. While we hope that our findings provide valuable insights for informing the design of future algorithms and benchmark metrics, our focus is limited to initiating a discussion instead of providing a complete solution.
> We will broaden our discussion and include details on conclusions that should not be drawn from our observations (e.g. relating to use-cases such as position control). We are happy to discuss in case the reviewer has further directions in mind.
>
>
> >I find Figure 1 somewhat overwhelming. [...] I'd find a useful summary.
>
> Figure 1 shows learning curves to provide insights into the exploratory learning phase as well as the final policy performance. We will follow the reviewer’s suggestion to expand the discussion and will also add summary metrics indicating the performance mean and standard deviation, with their respective standard deviations, aggregated over all 8 environments. For example, Gaussian MPO yields (849 +/- 104) +/- (62 +/- 113), while Bernoulli MPO yields (838 +/- 92) +/- (25 +/- 21). We will furthermore add an enlarged version of Figure 7 to the appendix for better visibility.
>
>
> >I am not quite convinced about the exploration section. It is almost by design that bang-bang spans a bigger region (which the authors mention). [...] I'd appreciate algorithmic developments towards increasing exploration in the Gaussian setting.
>
> We agree that exploration in Gaussian policies is a very important topic, particularly deep exploration in the face of sparse reward feedback. The simple Bang-Bang controller provided superior exploration under sparse rewards in Figure 10, while a common approach to mitigating bang-bang behavior in Gaussians - action penalties - proved to hurt exploration. We therefore hope that these insights may help to inform future exploration approaches for Gaussian policies.
>
> We thank the reviewer for referencing the SOP algorithm that ensures exploration in SAC without an entropy objective. We will include this in our discussion of related works, as well as the linked case study from the “Learning to Run” competition.
>
>
> >The authors do not specify technical details of the environments. It'd be nice to make the paper self-contained. [...] I'd appreciate more details about discretized SAC than Alg 2 (say what is the bijector).
>
> We will expand on the environment descriptions in the Appendix and will provide task objectives as well as input/output dimensions for each domain. We will furthermore add details on the bijectors, which includes tanh squashing for the Gaussian distribution and a matrix multiplication to recover class labels from one-hot vectors for the Bernoulli distribution. Both are then followed by a shift and scale operation to project the samples into the action range.
>
>
> We thank the reviewer for their insights and suggestions, and hope that we were able to address any open questions. We invite the reviewer to reconsider our submission based on the additional discussion, and to potentially adjust their score.

---

### Official Review · Reviewer_qvNS · 2021-07-15

**Rating:** 7
**Confidence:** 3

**Summary:**

The work investigates emergence of bang-bang control policies in
continuous RL problems, by making the connection to optimal control.
Exmpirically, it shows how continuous control problems are solved with
bang-bang policies, using 4 different algorithms, and that for most of
the domains in these experiments, a continuous action space is not
needed.
The experiments also explore modified reward functions designed to
increase smootheness of the learnt policies and their effect on the
(original) return obtained during exploitation.


**Ethical Concerns:**

I have found no ethical issues with the paper.


**Limitations And Societal Impact:**

Limitations and societal impact are adequately addressed.


**Main Review:**

Originality: the main contribution of the work is not a new method but
rather a new investigation of existing techniques about emergence of
bang-bang policies, with the goal to provide insights into designing
RL experiments and rewards. Previous work (that is also referenced)
focuses on approaches helping to avoid such policies but there is no
previous investigation of the kind presented in this work.

Quality: To my understanding the work is technicaly sound and supports
the claims, though is missing some details that would be good to
see. It consists of a theoretical analysis that provides an
explanation for emergence of bang-bang control, though without
complete proofs in the main paper. The experimental part would benefit
from more details (though parameters and some other details are
provided in the supplementary material): I may have missed it but
didn't see how often each experiment was run (or if each was only run
once). The work	presents a complete and carefully (and as far as I can
judge honestly) considered investigation. It does leave	open questions
for how to design rewards, benchmarks, or for how to avoid bang-bang
policies, and does not present a	solution to all	these, but rather
provides a basis for future work in these directions.

Clarity: The paper is clearly written and very well organised, and
presentation including graphs is of high quality.

Significance: I do believe the results are important, and the work is
set to be useful for future work in a number of directions - e.g., for
learning controllers where bang-bang policies a	better avoided like in
robotics, or potentially also for approaches speeding up
exploration. The work is not yet another paper improving
state-of-the-art on some benchmark (even though SOTA on some
benchmarks is achieved with the Bernoulli distributions used in the
experiments) but providing some new insights that I find more valuable
because of wider implications.


**Time Spent Reviewing:**

5

---

> ### Author Response · Authors · 2021-08-11
> **Response to Reviewer qvNS**
>
> We would like to thank the reviewer for their time and thoughtful comments. We appreciate them emphasizing the significance of potential implications that our findings can have on informing benchmark design and algorithmic trade-offs involving controller continuity and exploration abilities.
>
>
> >The experimental part would benefit from more details (though parameters and some other details are provided in the supplementary material): I may have missed it but didn't see how often each experiment was run (or if each was only run once).
>
> Training is performed on 4 seeds per run and evaluation under stochastic disturbances is performed on 5 rollouts per seed (see also Appendix C, L502-L504). We agree that stating this clearly in the main text will help to improve clarity and we will incorporate it accordingly.

---

### Official Review · Reviewer_wPsN · 2021-07-15

**Rating:** 6
**Confidence:** 3

**Summary:**

The authors analyze the emergence of implicit bang-bang (action saturation) control behavior in Deep RL. They claim three contributions, i) they argue that these are often actually optimal by making a connection to time-optimal solutions in optimal control theory, ii) they show empirically that the performance of an inherently bang-bang (discrete action) control policy yields comparable performance, and iii) they discuss possible extensions to reduce bang-bang behavior and their connection to exploration .


**Ethical Concerns:**

No.

**Limitations And Societal Impact:**

Yes.

**Main Review:**

The emergence of undesired bang-bang behavior can be a problem in real-world applications since it can damage mechanical systems or cause instability. Any insight into this problem would be valuable for control applications. Clarity of presentation is adequate and the paper contains a wealth of empirical experiments which appear to be carefully documented.

However, I have some concerns about the quality of the three main contributions as outlined below:
1) The first contribution is a proof from optimal control theory that bang-bang is optimal in continuous-time minimum-state problems without action penalties. The theoretical contribution here seems rather minor since it appears to follow directly from a book on optimal control [23]. Anybody that has played around with MPC also won't be surprised that the controller will saturate any action limits if the action penalty is low enough.
However, shouldn't pure bang-bang technically only be *optimal* for continuous-time, where max control output can be countered by an arbitrary small time step? For discrete-time, if the state is close enough to the target, the optimal action could be anywhere in-between the action limits. The authors claim that this continuous-time behavior translates to discrete-time systems because the time steps in RL are generally small enough, but this is never properly motivated. Even without examining this, running at higher Hz could of course be a solution, but it requires more compute and that the state is also available at that rate.
 The paper does raise a valid point about the importance of including action penalties in benchmark design, otherwise the agent will certainly saturate the action constraints, which may be undesirable in practice. Although the authors use the DeepMind control benchmarks as examples here, to the best of my knowledge, all the gym Mujoco benchmarks actually do include some kind of action penalty. I don't know if they are suitable or not. It also seems unlikely that bang-bang behavior observed in applications is just because they forgot to include an action penalty. This avenue is never really explored, and the paper never precisely defines what type of bang-bang behavior they are trying to fix - is it all kinds of action saturation, or just quick oscillations near an equilibrium? The latter may sometimes actually be desirable, but the latter never is.

2) The second contribution is empirical experiments showing that a discrete-action (bang-bang) control policy achieves performance comparable to regular continuous Gaussian ones. However, this just proves that current methods with Gaussian policies perform as poorly as a bang-bang controller. If the Gaussian ones often exhibit bang-bang behavior as purported in the motivation to this paper, this result does not appear particularly surprising. Finally, this focuses on training curves, which is not the same thing as the final performance. The tracking error near an equlibrium will often be dominated by the penalties to get there, especially if a square norm is used. It is unfortunate that they never mathematically define what constitutes harmful bang-bang behavior. The only attempt at measuring this are the action distribution plots in Figure 2, but it is difficult to see what is spurious oscillation and action saturation due to design of the reward function (low action penalty).

3) The third contribution is a purported discussion of possible extension to reduce bang-bang behavior by including action penalties. I didn't find anything supporting this except the recommendation to introduce action penalties. An important observation, but I suspect many practitioners with control background were already aware of this.
The authors also argue that action penalties can impede exploration and try to disentangle these. I did not find this line of reasoning entirely convincing. Couldn't changing the action penalties also drastically change the optimal strategy (e.g. run vs. walk cycles are quite different), so even if the lack of action penalties frees it up to visit more of the state space, it may be learning in the wrong region of policy space (e.g. you really wanted an energy efficient walk, not a run)? Are plots comparing state coverage meaningful here?

In conclusion, the authors target an important problem but the contributions appear minor or vague. The main finding only seems to pertain to the benchmarks in the DeepMind control suite that lack action penalties. The paper fails to convince me that a simple lack of action penalties is the main reason why DeepRL policies sometimes exhibit harmful oscillations, in the real world or even in other common control benchmarks.

Some details:

L18-54 - Intro and motivation is a bit fuzzy with several claimed minor contributions

L91: "The continuous time-setting simplifies analysis and provides a good approximation under the high sampling rates that are common on continuous control benchmarks and real-world robotic system." - One would rather see the discrete time as an approximation to continuous time when it comes to control tasks.

L112: "Nevertheless, we find that in this work the discretizaton intervals are small enough to not affect optimality of the learned policies." - based on what?

**Time Spent Reviewing:**

5

---

> ### Author Response · Authors · 2021-08-11
> **Response to Reviewer wPsN**
>
> We would like to thank the reviewer for their time and insightful comments. We are excited that they recognize the importance of studying emergent bang-bang behavior in RL policies and agree that insights into this phenomenon are valuable in light of (real-world) controls applications. In the following, we will try to clarify open questions and address remaining concerns.
>
> **Summary of contribution**:\
> Our contribution focuses on understanding agent behaviour, in particular with respect to Bang-Bang control and Bernoulli policies, and point towards potential challenges of current performance metrics in RL benchmarking. From a policy optimization perspective it is surprising that Bernoulli distributions enable stable learning and that continuous action spaces offered by Gaussians are not integral to current state-of-the-art performance. This is particularly striking as the original algorithms were designed with Gaussian policy heads and hyperparameters were only minimally adjusted. We provide both theoretical and empirical intuition for the success of Bang-Bang control. We also discuss pitfalls when trying to prevent this kind of behaviour by regularizing Gaussian policies via action penalties which can induce local optima and exploration challenges. Overall, these findings are meant to provide more subtle insights into the behavior exhibited by trained agents and to inform potential adjustments to evaluating agent performance.
>
> **Responses**:
> >The theoretical contribution here seems rather minor since it appears to follow directly from a book on optimal control [23].
>
> Our theoretical derivations of Bang-(Off-)Bang control primarily serve to bridge the optimal control perspective with the evaluated RL settings. The theory section therefore serves as a motivation and provides a theoretical perspective on our surprising empirical findings.
>
>
> >The authors claim that this continuous-time behavior translates to discrete-time systems because the time steps in RL are generally small enough, but this is never properly motivated. [...] running at higher Hz could of course be a solution, but it requires more compute and that the state is also available at that rate.
>
> The environments considered here generally run the control loop at 50-100Hz. Real-world systems may even run control at a higher rate, e.g. Boston Dynamics’ Atlas at 166Hz in [1] and ANYbotics’ ANYmal at 100Hz in [2]. We briefly investigate the effects of down-sampling the control frequency in Figure 4 (right, “Frequency”) and do not observe significant differences between the Gaussian and Bernoulli policies. We completely agree that control bandwidth can be a bottleneck for low-cost real-world systems, but view our contribution as pointing towards potential shortcomings of current performance metrics in RL benchmarking while highlighting the unexpected efficiency of learning with simple Bernoulli policies.
>
>
> >It also seems unlikely that bang-bang behavior observed in applications is just because they forgot to include an action penalty. This avenue is never really explored, and the paper never precisely defines what type of bang-bang behavior they are trying to fix - is it all kinds of action saturation, or just quick oscillations near an equilibrium? The [former] may sometimes actually be desirable, but the latter never is.
>
> We are primarily concerned about quick oscillations and extended application of maximum magnitude actions, though the latter can be difficult to disentangle from the desired behavior encoded by the reward function. For example, on the nominal Walker task with action limits [-1, +1] we find that in a trained Gaussian agent 36% +/- 4% of all action deltas are larger than 1.0 and 15% +/- 2% of all action deltas are larger than 1.9. Furthermore, 24% +/- 6% of all actions are part of sequences that maintain at least 0.9 action magnitude for more than 10 consecutive steps, and 5% +/- 3% for more than 20 consecutive steps. We will expand this discussion in the manuscript and include the corresponding metrics. Overall, these findings are meant to provide more subtle insights into the behavior exhibited by trained agents than only performance scores and to inform potential adjustments to evaluating and measuring agent performance.
>
>
> >It is unfortunate that they never mathematically define what constitutes harmful bang-bang behavior. The only attempt at measuring this are the action distribution plots in Figure 2, but it is difficult to see what is spurious oscillation and action saturation due to design of the reward function (low action penalty).
>
> We will include the above discussion and the suggested additional metrics relating to switching frequencies and maximum load intervals for improved clarity.
>
>
> >If the Gaussian ones often exhibit bang-bang behavior [..., the] result [that Bernoulli policies achieve comparable performance] does not appear particularly surprising.
>
> We agree that performance similarity of distinct policy types that converge to similar action distributions is not necessarily surprising (although the distribution similarity at convergence is not common knowledge in the RL community). However, from a policy optimization perspective it is surprising that Bernoulli distributions enable stable learning and may even improve performance on some tasks. This is particularly striking as the underlying algorithms were designed with Gaussian policy heads and the underlying hyperparameters were only minimally adjusted. Furthermore, while the Gaussian policies may leverage continuous actions throughout learning, their rich continuous action space is not integral to achieving strong performance on these common RL benchmarks.
>
>
> >The third contribution is a purported discussion of possible extensions to reduce bang-bang behavior by including action penalties. I didn't find anything supporting this except the recommendation to introduce action penalties.
>
> Our discussion focuses on the established approach of using action penalties and we consider regularizing the actions’ absolute value, square value, or change in value. We do not recommend the introduction of action penalties, and instead highlight potential drawbacks that this established method has in the context of RL. Particularly in sparse reward settings, action penalties can have a detrimental effect on exploration and give rise to complex trade-offs. The Bernoulli policies outperformed the Gaussian policies on the sparse tasks in Figure 10, highlighting the need for regularization strategies that do not interfere with deep exploration. We will make this more explicit in the manuscript and reformulate our third contribution to “We discuss the introduction of action penalties as a common method to reduce the emergence …”.
>
>
> >Couldn't changing the action penalties also drastically change the optimal strategy [...], so even if the lack of action penalties frees it up to visit more of the state space, it may be learning in the wrong region of policy space [...]? Are plots comparing state coverage meaningful here?
>
> We agree and this further supports our point that in RL settings it is not as simple as adding arbitrary action penalties to regularize the controller. Their presence may require reward engineering to prevent undesirable local optima that affect the character of learned behaviors. We provide and instance of this in Figure 7, where a quadratic penalty on the Walker task leads the Gaussian to learn a regularized policy that maintains most of the original performance as measured by the nominal reward function (middle plot, “No action cost”), while strongly reducing nominal performance on the Quadruped task. The state coverage plots in Figure 5 visualize general exploration advantages that Bernoulli policies can have over Gaussian policies and provide an intuition for the strong performance displayed on the sparse tasks presented in Figure 10.
>
>
> >L112: "Nevertheless, we find that in this work the discretization intervals are small enough to not affect optimality of the learned policies." - based on what?
>
> We will reformulate this to “... small enough to yield very strong performance with purely bang-bang policies.” These observations are based on empirical results relative to state-of-the-art Gaussian agents, particularly the results in Figure 1.
>
>
> >The paper fails to convince me that a simple lack of action penalties is the main reason why DeepRL policies sometimes exhibit harmful oscillations.
>
> As mentioned above, we agree and show in our paper that it is in fact not as simple as introducing action penalties, as this can introduce subtle trade-offs relating to local optima (e.g. Figure 7) and exploration capabilities (e.g. Figure 10). Instead, we show that across several recent RL algorithms benchmarking performance of normal Gaussian policies can be matched by simple Bernoulli policies and provide intuitions for the underlying reasons (e.g. reward structure, characteristics of converged Gaussians, exploration capabilities). This is surprising, as research in continuous RL primarily relies on Gaussian policies with many algorithms designed around Gaussian distributions. Our goal is primarily to initiate a discussion in the community and we hope that our insights can inform design of future algorithms and metrics used for benchmarking performance in continuous control RL.
>
>
> We thank the reviewer for their extensive feedback, and hope that we were able to clarify any open questions and address remaining concerns. We invite the reviewer to reconsider our submission based on the additional discussion and clarifications, and to potentially adjust their score.
>
> 1. Johnson, Matthew, et al. "Team IHMC's lessons learned from the DARPA robotics challenge trials." Journal of Field Robotics 32.2 (2015): 192-208.
> 2. Lee, Joonho, et al. "Learning quadrupedal locomotion over challenging terrain." Science robotics 5.47 (2020).

---

> > ### Comment · Reviewer_wPsN · 2021-08-27
> > **Thanks for clarifications.**
> >
> > The authors addressed enough of my concerns to raise my score. The problem is important and the experiments are valuable. The paper will be stronger with the suggested clarifications.

---

### Official Review · Reviewer_48Th · 2021-07-18

**Rating:** 8
**Confidence:** 5

**Summary:**

The paper deals with the question namely the issue when an optimal
continuous ("soft") control can be superseded/replaced or even needs
to be implemented by a Bang-Bang control.

**Limitations And Societal Impact:**

Limitations etc. are very well discussed throughout.

As for Societal Impact, this is essentially basic research; there should be an N/A option on the "Societal Impact" statement. Asking authors of basic research to comment on societal impact, unless it is directly relevant is no better than asking them to read entrails.



**Main Review:**

The paper deals with an interesting question, namely the nature of
optimal control solutions and whether a simplistic bang-bang might
work just as well as a continuous (e.g. Gaussian) control. The reviewer
particularly likes the idea to ask what the nature of the problems
would be which would favour bang-bang, rather than just presenting
"yet another algorithm" - in this sense, this paper is researching the
*nature* of problems leading to particular optimal effects. The paper
also is careful to note that RL entangles exploration, learning and
execution, and the issue of trade-offs. Amongst other, action costs
may discourage bang-bang, but for Gaussian exploration they may
confine the exploration; it is clear that there are many subtleties
that dictate the preferred outcome.

A very compact derivation (based on Pontryagin) and discussion of
emergence of bang-bang and a set of examples where bang-bang is
predominantly seen in a number of "walking"-type scenarios.

A number of instructive analyses demonstrate the prevalence of various
solutions and dynamics under varying action costs and thus balances
and thus sheds light on the nature of problems and cost structures
that favour bang-bang against soft controls. An interesting research
paper on the meta-question of what solutions one can expect for
certain problem formulations.

I have no particular comments on this well-written paper.


**Time Spent Reviewing:**

2h 15m

---

> ### Author Response · Authors · 2021-08-11
> **Response to Reviewer 48Th**
>
> We would like to thank the reviewer for their time and positive feedback. We are elated about their appreciation of research that aims at better understanding characteristics and limitations of problem formulations and performance metrics currently used in RL research. We hope that our insights will help to inform future agent developments and design of relevant comparative metrics as well as benchmarks.

---

### Decision · Program_Chairs · 2021-09-27

**Decision:**

Accept (Poster)

**Comment:**

After reading each other's reviews and the authors' feedback, the reviewers discussed the merits and flaws of the paper.
The authors' answers have been appreciated by the reviewer, who reached a consensus about accepting this paper.
I want to congratulate the authors and invite them to modify their paper following the reviewers' suggestions.